# Exosomes and Other Extracellular Vesicles with High Therapeutic Potential: Their Applications in Oncology, Neurology, and Dermatology

**DOI:** 10.3390/molecules27041303

**Published:** 2022-02-15

**Authors:** Urszula Szwedowicz, Zofia Łapińska, Agnieszka Gajewska-Naryniecka, Anna Choromańska

**Affiliations:** Department of Molecular and Cellular Biology, Wroclaw Medical University, Borowska 211A, 50-556 Wroclaw, Poland; zofia.lapinska@student.umed.wroc.pl (Z.Ł.); agnieszka.gajewska-naryniecka@umed.wroc.pl (A.G.-N.)

**Keywords:** extracellular vesicles, EVs, exosomes, drug delivery

## Abstract

Until thirty years ago, it was believed that extracellular vesicles (EVs) were used to remove unnecessary compounds from the cell. Today, we know about their enormous potential in diagnosing and treating various diseases. EVs are essential mediators of intercellular communication, enabling the functional transfer of bioactive molecules from one cell to another. Compared to laboratory-created drug nanocarriers, they are stable in physiological conditions. Furthermore, they are less immunogenic and cytotoxic compared to polymerized vectors. Finally, EVs can transfer cargo to particular cells due to their membrane proteins and lipids, which can implement them to specific receptors in the target cells. Recently, new strategies to produce ad hoc exosomes have been devised. Cells delivering exosomes have been genetically engineered to overexpress particular macromolecules, or transformed to release exosomes with appropriate targeting molecules. In this way, we can say tailor-made therapeutic EVs are created. Nevertheless, there are significant difficulties to solve during the application of EVs as drug-delivery agents in the clinic. This review explores the diversity of EVs and the potential therapeutic options for exosomes as natural drug-delivery vehicles in oncology, neurology, and dermatology. It also reflects future challenges in clinical translation.

## 1. Introduction

In the last two decades, interest in investigating new nanoparticulate delivery systems has grown expeditiously. The main goal is the improvement of pharmacokinetic and pharmacodynamic therapeutics.

Extracellular vesicles (EVs) are essential carriers of intercellular communication in both prokaryotes and eukaryotes, conditioned by the active molecules they carry, such as proteins, lipids, and nucleic acids. The unique content of EVs can deliver many different transmitters, even to areas distant from the donor cell [1,2].

EVs, and above all exosomes, are often classified by researchers as Smart Drug-Delivery Systems (SDDS), i.e., small molecules that can carry and release biologically active compounds in a controlled manner [3]. Similarly to other representatives of SDDSs, they use specific properties of pathological target cells, such as increased permeability of their cell membrane or overexpression of some receptors, to operate with high precision [4]. Examples include micelles, liposomes, and dendrimers, but also gold nanoparticles and mesoporous silica nanoparticles [5]. When assessing their suitability as a safe carrier, their biodistribution, chemical stability, toxicity, and ability to accumulate in internal organs are taken into account. Exosomes perform particularly well compared to others in terms of biosafety and immunogenicity and the diversity of the cargo introduced into them. As natural and therefore diverse drug-delivery carriers, they do not need any special modifications to be used safely. For the other SDDS representatives, physicochemical modifications are necessary to achieve adequate stability and performance. Their size and shape are also of particular importance, because slight deviations can cause a dramatic change in their properties. Exosomes, although so readily used in, e.g., neurology and various pathological changes in the head and brain area due to their ability to cross the blood-brain barrier, have some limitations in use in other places, such as adipose tissue, skeletal muscles, or the heart [6]. Nevertheless, it has not yet been possible to artificially create a better delivery system for various compounds capable of performing similarly to complex native exosomes.

Numerous studies show that EV membranes’ composition, content, and size are heterogeneous and dynamic. Due to their structure and properties, EVs quickly found application in a wide range of clinical disorders. This comprehensive review focuses on current knowledge about EVs and their role in enhancing treatment of the diseases most threatening to people in these times.

## 2. The EVs Classification and Theoretical Background

Extracellular vesicles (EVs) are signaling-correlated phospholipid bilayer-delimited particles secreted by almost all types of cells (both prokaryotic and eukaryotic) under physiological and pathophysiological conditions [7]. Their presence is noted in culture supernatants and biological fluids, such as saliva, urine, blood, breast milk, pleural ascites, and cerebrospinal fluids [8,9,10] According to the size and the biogenesis pathway, EVs can be distinguished into three main groups: exosomes, ectosomes, and apoptotic bodies. Each of these is described below.

### 2.1. Exosomes

Exosomes (Exos), also referred to as intraluminal vesicles (ILVs) with the size range 30–150 nm, are produced as a result of inward budding of the endosomal membrane during the maturation of multivesicular endosomes (MVEs) [11]. Their production is based on the endolysosomal pathway, during which Exos form as intraluminal vesicles (ILVs) within MVEs [12]. Exos are released by the cell in the aftermath of MVE fusion with the membrane. Endosomal sorting complexes required for transport (ESCRT) machinery, Rab proteins, CD36, and sphingolipid ceramides have been shown to play an essential role in Exos biogenesis [12,13,14,15,16,17]. The dependencies between these elements and their exact position have been described by Willms et al. and Zhang et al. [12,18].

### 2.2. Ectosomes

Ectosomes, also called microvesicles (MVs) or microparticles, range from 100 to 1000 nm [19]. Together with apoptotic bodies, they belong to the EV subtype, created during pinching off the plasma membrane [11]. MVs are secreted from the cell as a result of its contraction and peeling in unique spaces. The exact molecular aspects of the MV biosynthesis route are still poorly understood. Based on the available literature, it is thought that cytoskeleton components, i.e., actin fibers and microtubules together with molecular motors and SNAREs, are required for this process [20,21]. Interestingly, it was pointed out that the MV uptake by recipient cells is an energy-dependent activity, and it is inhibited in lower temperatures [22].

### 2.3. Apoptotic Bodies (ApoBDs)

Apoptotic bodies (ApoBDs), also known as ‘apoptotic blebs’ or ‘apoptotic vesicles,’ represent the largest subpopulation of EVs, with sizes ranging from 50 nm–5 μm, close to the size of platelets. ApoBDs are released by the cell membrane blebbing, or fragmenting, when it is compelled to undergo apoptosis [23]. The membrane divides the degraded cell into small apoptotic bodies, which can then be absorbed and digested by phagocytic cells. These molecules carry fragmented apoptotic DNA and cellular organelles, i.e., endoplasmic reticulum or mitochondria [19]. ApoBDs were first considered as cell debris and not consulted during preliminary EVs investigations. Kakarla et al., based on current research, mentioned that apoptotic bodies with a size less than 1 µm have been recluded as an EV subtype called apoptotic microvesicles (ApoMVs), mainly due to the fact of their membranes’ superior integrity for molecular exchange.

## 3. The Isolation and Loading of EVs

The key factor in selecting a method for preparing extracellular vesicles (EVs) is primarily their destination. This is what the method of isolating the EVs depends on. Of course, it is important to obtain a good-quality product without the non-exosomal impurities, but it is even more desirable to isolate the intact particles—undamaged externally, and without changes in the quality of the internal cargo.

The most common methods used to isolate EVs are ultracentrifugation and ultrafiltration—procedures based on the physicochemical properties of EVs [23]. These methods are favored mainly for the simplicity of the insulation and the quantity of the final product, but unfortunately, the quality and purity of the isolated vesicles are not satisfactory [24]. Answers to this problem can be found in techniques such as size-exclusion chromatography, gravity-driven filtration, or gradient ultracentrifugation, which obtain safer products, free from bacteria or viruses [25]. However, the above methods, though very popular, unfortunately are not the best choice when it comes to producing EVs for use as a drug-delivery system. For this purpose, scientists have recently indicated two main methods that seem promising: microfluidic technologies and hydrostatic dialysis [26]. Both allow the EVs to be isolated without affecting fragile internal cargo structures, such as RNAs or other proteins. During the relatively rapid process itself, there are no major losses [4].

Another important aspect is the encapsulation of chemotherapeutic drugs and its efficiency. There are two kinds of methods for loading EVs [23]. The first is based on isolating EVs that already have the desired cargo, and consists in targeting the cell from which they are derived. The second method focuses on acting on the obtained EVs. In the case of the production of EVs for therapeutic purposes, it is recommended to use the second option, since it is easier to control the quality and quantity of a particular substance introduced into EVs. The vast majority of these techniques are based on increasing the permeability of the lipid bilayer membrane and smooth delivery of the drug inside [27]. Particularly noteworthy are electroporation, sonication and saponin-mediated permeabilization. When used with a suitable membrane stabilizer, it is possible to restore the original EV structure without the formation of impurities, such as aggregates, and no vesicle fusion occurs [28].

Currently, it is difficult to indicate the best method of obtaining EVs. Their complexity and diversity make them unique, but also raise many questions and unknowns. The only thing that can be safely said is that each research plan must be approached individually. More research is needed to determine the best solution.

## 4. Clinical Use of EVs

Based on proteomic studies of extracellular vesicles (EVs) released by cells cultured in vitro, tissue culture cells, or EVs isolated from body fluids, catalogues of proteins present in several types of EVs were created. They are specific markers reflecting the cellular origin and the secretion mechanism of the bottom microbubble [29]. EVs primarily contains cytoskeleton and plasma membrane proteins, cytosolic proteins, heat shock proteins, and vesicle transport proteins. The proteomic profiles of EVs largely depend on their isolation method. These include differential ultracentrifugation, density gradient centrifugation, filtration, and exclusion chromatography. Since the differences in EV physical properties and composition are small, it is difficult to distinguish between the different subgroups of EVs after their isolation. Moreover, the same type of cells may secrete different subgroups of EVs depending on environmental factors, and the protein content in the same EV subgroups depends on the activating stimulus [30]. The markers of EVs include tetraspanins (CD9, CD63, CD81, and CD82), major histocompatibility complex (MHC) molecules, 14-3-3 proteins, heat shock proteins, Tsg101, and ESCRT-3 complex [31]. However, to establish whether there are specific markers associated only with particular subgroups of EVs, it is necessary to establish a consensus regarding the methods of isolating the analyzed microbubbles [30].

Based on the lectin analysis of EV microarrays derived from T lymphocytes, it was found that their glyco-pattern differs from the parent cell membrane [32]. Studies have shown a high level of mannosylated epitopes within EVs compared to the stem cell membranes of human lymphocyte, melanoma, and colorectal cancer cell lines. Changes in the glycosylation patterns of EVs in pathological states have also been reported [33]. It was also observed that EVs derived from B lymphocytes are enriched in sialic acid with α-2,3 linkages, which enables their uptake by CD169 molecules on macrophages [34]. Several studies confirm that glyco-interactions are essential in the sorting and interacting of EVs on target cells. Moreover, surface glycosylation patterns are critical to the uptake of EVs by recipient cells [35].

The similar protein–lipid composition of different types of EVs may result from the mechanism of membrane component sorting, which is conditioned by the membrane curvature. The membrane components have a certain freedom of movement and arrange themselves in the most energetically favorable conformations [36]. Thus, they determine the local composition of the membrane and its curvature. Curvature-based sorting of proteins and lipids was demonstrated in an artificial membrane model and eukaryotic membranes [2]. The mechanism of sorting the curvature of membrane components is induced in the stem cell during membrane budding, and it is mainly responsible for the size, shape, and composition of the formed EVs. The mechanism concerns vesicles formed inside the MVB or by budding from the plasma membrane [37]. Certain structural elements have been shown to promote the formation of membrane curvature. These include proteins containing the BAR domain (Bin/Amphiphysin/Rvs), which induce tubular and alveolar membrane structures [38]. ESCRT proteins play an essential role in the cleavage of membrane germs. The ESCRT complex recruits exosome components by binding to ubiquitinated proteins [39]. The protein composition of EVs determines their functionality in several different ways. EVs can induce intracellular signaling pathways by simply interacting with surface receptors or ligands of target cells, or by being internalized [2].

The level of unbound EVs in the body reflects the balance between their generation and their binding to the target cell. Independent studies suggest that the half-lives of exogenous EVs are short, but that differences are observed depending on their origin. Rabbit biotinylated EVs were cleared entirely from the rabbit’s circulation within 10 min [40], while EVs from red blood cells and B16 melanoma cells showed a clearance of more than 90% after half an hour [41]. EVs from human platelet concentrate remained in circulation for up to 5.5 h [42]. The biodistribution of EVs depends on the source of the stem cell and the availability of different types of target cells to internalize circulating EVs. It is worth noting that both the interaction and the uptake of EVs by cell surfaces are regulated by the mutual expression of intercellular adhesion molecules such as the ICAM-1 and LFA-1 integrin [43].

The interaction of EVs with target cells occurs through ligand-receptor interactions. The proteins MHC I, MHC II, and tetraspanins present on the surface of EVs induce different signaling pathways of target cells [44]. Proteins from the HSP family (HSP 27, 60, 70, and 90) undergo some intracellular redistribution to the cell membrane, lipid rafts, or MVB, in pathological states, such as cancer. They successively find their place in EVs, through which they interact with the membrane receptors of distant target cells, propagating the pathological condition [45]. EVs also contain active lipolytic moieties, leading to the formation of lipid mediators (fatty acids and prostaglandins) interacting with peripheral G-protein coupled receptors and nuclear receptors in target cells. An example of the functional role of EV ligands for membrane receptors in the presence of ligands for death receptors in EVs. Human NK cells have been shown to release EVs that actively induce target cell death. In addition, human tumor cells release EVs containing ligands for death receptors. The EVs released by tumor cells, including the FasL and TRAIL ligands, may be involved in the lysis of lymphocytes. As a result, death is not induced within cancer stem cells that release EV [46].

In addition to mediating the exchange of intercellular information, EVs are carriers of soluble mediators, including cytokines. They are released into the extracellular space after ATP binds to the P2X7R receptor on EV [47]. Cytokines localized within EVs include IL-1β, IL-1α, IL-18, macrophage migration inhibitory factor (MIF), and IL-32 [48]. Platelets have been shown to release EVs containing vascular endothelial growth factor (VEGF). Interestingly, the VEGF present in EVs derived from tumor cells is released in a bioactive form only in the presence of the acidic pH characteristic of the tumor microenvironment [49].

Extracellular RNA molecules can be encapsulated in EVs. The presence of functional RNA in EVs was first described in 2006 [50]. EVs can contain intact mRNAs, mRNA fragments, miRNAs, long non-coding RNAs, small non-coding RNAs, ribosomal RNAs, and tRNA fragments. Studies have shown that the RNA present in EVs can compete with cellular RNA for binding to miRNA in recipient cells [51]. The release of specific RNA molecules may also influence the regulation of gene expression in parental cells. The studies indicate a particular repertoire of miRNAs selectively exported to EVs, which shows an active mechanism of RNA sorting into EVs [52]. The presence of DNA particles has also been demonstrated in EVs. The observed forms of DNA include mitochondrial (mtDNA), single-stranded, and double-stranded [53]. EVs may constitute the pathway by which altered mtDNAs derived from cancer cells enter other cells, promoting the diffusion of pathological changes. Genomic dsDNA molecules reflecting the mutation state of cancer stem cells have been observed in EVs released from the above cells [54]. The physiological significance of DNA charge in EVs is still poorly understood [2].

Loading cargoes into EVs necessitates overcoming the barrier of the EV membrane. The strategies to encapsulate molecules exogenously into EVs can be categorized as passive or active [55]. Electroporation is a well-known method for passive packaging. The appropriate electric voltage applied to the surface of the membrane causes so-called electrophoresis, which provides additional transport paths, e.g., for drugs that have difficulty penetrating the cell membrane. Electroporation is used to encapsulate all kinds of cargoes, including proteins and mRNAs. Active modification strategies include genetic engineering and chemical modification. The gene sequence of a guiding protein or polypeptide is fused with a selected EV membrane protein through genetic EVs engineering. Donor cells are transfected with plasmids encoding the fusion proteins. LAMP-2B protein is often used to display a targeting motif. It has been observed that the N-terminus of LAMP-2B is displayed on the surface of exosomes. It can also be appended with targeting sequences [56]. The chemical modification allows an extended range of ligands to be displayed to EVs via conjugation reactions. Conjugation reactions can modify EV surface proteins, but such a reaction often lacks control of site-specificity. Covalent modification may damage the structure and function of the vesicle. Lipids or amphipathic molecules can also be inserted into the lipid bilayer of EVs, releasing their hydrophilic parts to be arranged on the surface. However, this method may also increase the toxicity of exosomes [55].

As presented above, EVs as carriers for a wide range of molecules may serve as a novel potential tool in a broad scope of therapeutic solutions and diagnostic methods, such as anti-tumor therapy, immune-modulatory, and drug delivery. The authors of this publication, at their discretion, focused on the three most interesting areas of modern medicine where EVs have the most promising application. Oncology, neurology, and dermatology are areas in which non-invasive and non-toxic drugs are used to play a crucial role in the therapeutic process for the patient. Acting on the molecular level reduces to a large extent, or even eliminates, the side effects of many commonly used therapies. EVs offer the possibility of curing neoplastic disease without the need for surgical intervention or the use of chemotherapy, which strain the human organism. Their ability to penetrate the nervous system without disturbing the integrity of the blood–brain barrier means bypassing the need for hazardous procedures, and offers the possibility of treating the cause of the problem in situ. Therapies with EVs are a step further than those with stem cells, because the carcinogenic potential is minimized and the regenerative benefits are at least the same and often much higher [57]. So far, no significant adverse systemic effects of EVs have been demonstrated to preclude their use, in contrast with steroids, which need to be constantly monitored for their side effects in dermatological diseases with an immunological background [58]. Numerous studies on animal models and clinical trials support the superiority of EV therapy in terms of safety and long-term therapeutic effects, which are the most important for patients with already burdened immune systems.

## 5. EVs and Cancer Immunotherapy and Vaccination

Cancer remains one of the leading causes of death worldwide. Each year, nearly 10 million people die from this deadly disease, and over 19 million new cases were recorded in 2020 [59]. Cancer therapies include surgery, chemotherapy, radiotherapy, and immunotherapy. Knowledge and understanding of tumorigenesis, the role of the tumor microenvironment in cancer progression, and immuno-oncology have raised and shed new light on the development of modern anticancer therapies. Cancer immunotherapy (CIT) is the approach that generally activates or modulates the patient’s immune system to fight cancer. So far, CIT covers immune checkpoint blockade (ICB) therapies, chimeric antigen receptor (CAR) T cells, and cancer vaccines [60,61,62]. Even with impressive advances in CIT, there are still issues to be overcome, such as limited response rate, autoimmune reactions, and cytokine release syndromes [63]. In addition, the effectiveness of immunotherapy differs among cancer types. One reason is that tumor immunogenicity varies among cancers of the same type and between different malignancies. Some cancers show a high immunogenicity level, while others do not [64]. The other possible cause is the immunosuppressive tumor microenvironment, which can inhibit anti-tumor immune responses. The tumor site comprises cancer cells and stromal cells, such as fibroblasts, immune cells, adipose tissue, blood vessel networks, and the extracellular matrix. Tumor cells communicate with their niche through EVs, and their role in cancer biology, immunology, and clinical diagnostics has been reviewed [65,66,67,68,69].

Exosomes have a unique molecular profile that depends on the host-cell origin. The capacity of exosomes to bring bioactive cargoes into the immune and cancer cells’ cytoplasm has sparked a strong interest in their application in cancer treatment and drug delivery. Advantages of exosomes for implication in therapeutics include high biocompatibility, stability, and small size, which allow for prolonged circulation, evasion of phagocytosis, and passive targeting of cancer because of vascular leakage of tumor vessels and disabled lymphatic drainage. They can also deliver their cargo directly to the cell cytoplasm and avoid lysosomal degradation. For immunotherapy, the feature of exosomes that outstrips immune cell-based therapy is that they are resistant to the suppressive effects of IL-10 and TGFb released by tumor cells.

Moreover, exosomes can deliver various signals simultaneously [70]. Over the past decade, exosomes have been extensively studied in oncology, autoimmune diseases, cardiology, and infections [71,72,73]. We present advancements in exosome-based studies of therapeutic applications in some cancers to show their potential for developing modern, effective treatments.

### 5.1. Pancreatic Cancer

Pancreatic ductal adenocarcinoma (PDAC) is one of the deadliest cancers, with a 5-year survival rate of 8% due to late-stage diagnosis, high metastatic potential, and insufficient response to the currently available treatments, including immunotherapy. Surgical resection remains the only hope for long-term survival among these patients [74]. There is an urgent need to develop modern, effective therapies against pancreatic cancer. PDAC immunotherapy using exosomes is still a relatively new area of study. Pancreatic cancers are low immunogenic tumors surrounded by a protective, immunosuppressive tumor microenvironment. During the initiation of pancreatic tumorigenesis, the cancer is infiltered mainly through Treg cells, tumor-associated macrophages, and myeloid-derived suppressor cells (MDSC) [75]. This obstacle needs to be overcome for effective immunotherapy, and exosomes gained the interest of oncologists as a drug-delivery system. Exosomes with their high biocompatibility and ability to transfer functional molecules are promising tools for delivering chemotherapeutics, gene therapies, and immunotherapeutic drugs to cancer cells [76,77]. Some studies show the potential of exosomes as a delivery system for anticancer drugs and gene delivery for pancreatic cancer treatment. KrasG12D is a common mutation of KRAS GTPase in pancreatic cancer involved in tumor initiation, progression, and metastasis, but applying it as a therapeutic agent is challenging. Kamerkar et al. demonstrated an approach that uses engineered exosomes from normal fibroblast-like mesenchymal cells to deliver siRNA or shRNA specific to oncogenic KrasG12D (iExosomes). The therapy showed promising outcomes in early-treated mice, with tumor burden nearly undetectable and survival after 200 days. The treatment of mice in advanced cancer was less effective, prolonging life-expectancy but not inhibiting the progression of the disease [78]. The study brings hope for pancreatic cancer patients for new effective treatments, but further studies need to be conducted to improve the overall outcome and apply it to humans. Recent studies showed that the use of exosomes derived from NK cells could be a promising therapeutic approach in pancreatic cancer. Sun et al. demonstrated that NK cells (NK EVs) are capable of inhibiting the malignant transformation of co-cultured pancreatic cancer cells (PC) both in vitro and in vivo, via miR-3607-3p enriched exosomes targeting IL-26 [79]. The presented study showed that treatment with NK EVs inhibited cell viability, proliferation, and migration in vitro. There is a need for further research to check the capability of NK EVs treatment to inhibit tumor growth in animal models and humans. Immunotherapy that aims at the tumor microenvironment of PDAC was shown to be a promising strategy.

Zhou et al. [80] proposed a dual delivery system consisting of exosomes derived from bone-marrow mesenchymal stem cells (BM-MSCs), with electroporation-loaded galectin-9 siRNA subsequently modified with oxaliplatin (OXA) prodrug as an immunogenic cell-death (ICD) inducer. Prepared siRNA-EXO-OXA (iEXO-OXA) nanoparticles were shown to be preferentially delivered into tumor sites in PANC-O2 tumor-bearing mice after tail-vein injection. Randomly divided animals were vaccinated 5 times every 3 days with different therapeutics (OXA alone, EXO-OXA, iEXO, iEXO-OXA, gemcitabine, and as control group PBS). This research revealed that the CD8+ cytotoxic T lymphocyte population was the most abundant among mice treated with combined immunotherapy (iEXO-OXA); the lowest ratio of regulatory T lymphocytes was in the same group. This approach has the highest anti-tumor effect, with a notable reduction in tumor size and the longest life span compared to other treatments [80]. All these data together show that aiming not only for cancer cells, but also for immunosuppressive TME, is a good direction for modern immunotherapies. Unfortunately, the number of studies focusing on pancreatic carcinoma immunotherapy applying exosomes remains negligible, and currently, no clinical trials are being conducted.

### 5.2. Breast Cancer

Female breast cancer is the most commonly diagnosed cancer in women, with over 2 million new cases (11.7% of the total cases) and 684, 996 deaths (6.9% of the total cases) worldwide in 2020 [59]. There are three major subtypes of breast cancer, depending on the presence or absence of molecular markers for progesterone or estrogen receptors, and human epidermal growth factor 2 (ERBB2). Statistically, 70% of cancer patients show hormone-receptor-positive/ERBB2-negative phenotype, while 15% to 20% are ERBB2-positive and another 15% have triple-negative tumors (TNBC) missing all three standard molecular markers. Among all breast cancer subtypes, the triple-negative type is more likely to recur, and shows the lowest 5-year survival rate (85% for nonmetastatic triple negative cancer, 94% for hormone-receptor-positive, and 99% for ERBB2-positive cancer). In addition, ERBB2+ and TNBC show the highest risk of brain metastases [81]. Here we review a few papers showing that exosome-based platforms have the potential for new, more efficient therapies for breast cancer. A recently published study, which utilized mesenchymal stem-cell-derived exosomes as shuttlers of Taxol to the MDA-hyb1 breast cancer cell-bearing mice, showed they could efficiently target primary tumors and metastases, causing reduction in tumor growth and inhibiting metastases, with reduced side effects from the therapy [82]. Another study demonstrates the use of the synthetic multivalent antibodies retargeted exosome (SMART-Exo) platform designed by the group of Shi et al. [83] to work as an artificial modulator of cellular immunity, able to redirect immune effector cells and control their immunoreactivity. This platform was developed by them as a prospective novel targeted immunotherapy for HER-2-expressing breast cancer (ERBB2-positive). They prepared the genetic construct of anti-CD3-anti-HER2 bispecific scFv, and transfected the Expi293 cells derived from the HEK293 cell line to gain exosomes displaying bifunctional anti-antibodies on their surface (αCD3-αHER2 SMART-Exos). The data showed strong activation of T-lymphocytes mediated by αCD3-αHER2 SMART-Exos in an ERBB2-positive cancer cell-dependent manner in vitro, and their cytotoxicity against HER2-expressing breast cancer was CD3+T cells-dependent. The engineered exosomes showed minimal toxicity and immunogenicity, with no systemic cytotoxicity observed in animal experiments. Moreover, this preclinical study demonstrated that αCD3-αHER2 SMART-Exos administered to xenograft HCC 1954 tumor-bearing mice engrafted with human PBMCs resulted in significant tumor growth inhibition. The study showed that SMART-Exos loaded with anti-CD3-antiHER2 dual scFv antibodies concurrently recognized T cell surface CD3 and HER2 overexpressed by breast cancer cells, and promoted anticancer immunity in a specific and controlled way [83].

Further investigation and improvement of this approach may lead to effective therapy for breast cancer, giving hope for more prolonged survival for thousands of cancer patients. This exosome-based platform showed high potential for developing innovative immunotherapeutic therapies for robust and selective induction of specific anti-tumor immunity for not only breast cancer, but other cancer forms. Recently, Li et al. [84] presented a study identifying two subsets of adipose-derived mesenchymal stem cells (CD90^low^ADSCs and CD90^high^ADSCs) and showed that stimulation with LPS exosomes caused conversion of CD90^high^ADSCs to CD90^low^ADSCs, which demonstrated higher anticancer activity. Then they examined the anti-tumor activities of CD90^high^ and CD90^low^ ADSCs and ADSC-derived EVs in vitro and in vivo in an E0771 and 4T1, an immunocompetent syngeneic mouse model of breast cancer. In the next step, they used CD90^low^ADSCs-EVs as drug carriers for anti-oncogenic miRNA-16-5p to treat breast-cancer tumor-bearing mice. Researchers showed that exosomes derived from ADSCs with a lower level of CD90 expression slowed tumor-cell proliferation and lowered migration on a higher level than EVs from CD90^high^ADSCs in vitro assays. The in vivo experiments with the mouse model of breast cancer showed that CD90^low^ADSCs-EVs were inclined to slow tumor growth and reduce malignant mass significantly. The miR-19-5p mimic-loaded CD90^low^ADSCs-EVs showed robustly increased-level tumor-cell apoptosis, slowed tumor growth, and decreased tumor mass on a higher level than unmodified CD90^low^ADSCs exosomes [84].

The presented study revealed the potential of ADSC-derived exosomes as therapeutics for breast cancer. They also showed that the engineering of exosomes could be a promising approach to the anticancer efficacy of therapy. However, further studies must confirm the possible clinical application of the proposed treatment in an animal model and clinical trials. Metastatic TNBC is the leading cause of death from breast cancer due to the high rate of recurrence and metastasis after surgical intervention. Zhao et al. demonstrated novel engineered CBSA/siS100A4@Exosomes, which are biomimetic exosomes fabricated by coating cationic bovine serum albumin (CBSA)/siRNA(siS100A4) nano-complexes with the exosomal membrane of EVs derived from autologous breast cancer cells [85]. The CBSA/siS100A4@Exosomes were shown to be intensively taken up by mouse embryonic lung fibroblast in vitro, which was in line with in vivo distribution in the postoperative lung metastasis mice model, mainly in the liver, spleen, and lung on a higher level than liposome-based platforms. The mean number of metastatic nodules after treatment with CBSA/siS100A4@Exosomes was dramatically decreased compared to other therapies applied. Moreover, they demonstrated that biomimetic exosomes downregulated the expression of S100A4 and caused suppression of postoperative metastasis in TNBC. The Zhao group presented a promising approach to developing biomimetic exosome-based platforms to aim and eradicate cancer metastasis, an up-and-coming clinical approach for cancer therapy and prevention of tumor progression.

Reviewed studies showed that exosomes as natural nanoparticles have great potential for use in cancer therapy thanks to their natural features, biocompatibility, low toxicity, and targeting properties, but particularly due to the ease and feasibility of functionalization. The tables below present the literature data on the use of EVs in the treatment of neoplastic diseases presented in the text (Table 1 and Table 2).

### 5.3. Exosome-Based Cancer Vaccines

In contrast to vaccines for infectious diseases, cancer vaccines are therapeutic interventions. They can aim at various antigens expressed by tumor cells, including tumor-specific antigens (TSAs), such as mutated P53 and RAS, or tumor-associated antigens (TAAs), for example, MAGE-1 and HER2. Here we discuss the exosome-based vaccines, with a strong focus on dexosome-based (DEX) vaccines derived from dendric cells (DCs) and their potential in cancer therapy (Table 2). DCs, as professional antigen-presenting cells (APCs), play a crucial role in the regulation and initiation of innate and adaptive immune responses, and hold the unique capability of priming both naïve CD4 and CD8 T cells [87]. The dexosomes are EVs enriched with immunologically relevant components, including antigen-presenting MHC I/II molecules, costimulatory proteins (CD86, CD 40, CD80, CD1a-d), intracellular adhesion molecules (ICAM-1, CD54), integrins, and CD55 and CD59 molecules present on the DEX surface, allowing evasion of complement-mediated degradation. These features make these nanoparticles good candidates to trigger a CTL response against cancer. In 1996, Raposo et al. demonstrated that B lymphocytes secrete exosomes containing antigen-specific MHC class-II-restricted complexes, which trigger T cell responses [88]. It was the first documented evidence of the role of EVs in immunology. Two years after that, Zitvogel et al. showed that BM-DCs (bone-marrow-derived-DCs) pulsed with acid-eluted tumor peptides secrete DC-derived exosomes—dexosomes (DEXs) capable of mediating specific, long-lasting anti-tumor responses in tumor-bearing mice [89]. These findings paved the way for research in the field of exosome-based immunotherapy. The first clinical trials exploiting DEXs as a cancer vaccine were in 2005. The therapeutic method consisted of using autologous exosomes pulsed with MAGE-tumor antigens in patients with late stages of melanoma and patients with advanced non-small lung cancer (NSCLC). Both demonstrated that DEX-based anticancer vaccines are safe, with no severe side effects. However, no specific CD4+ or CD8+ T cell responses were detected in the peripheral blood of metastatic melanoma patients, and only 1/3 of NSCLS patients showed MAGE-specific T cell responses. The other 2/3 had increased NK lytic activity. These trials proved that the preparation of dexosome-based vaccines is clinically feasible from patient-derived dendric cells [90,91]. The work of Gehrman et al. showed that in treatment with a-galactosylceramide (aGC)/antigen ovalbumin (OVA)-loaded exosomes (Exo(aGC-OVA) derived from bone marrow dendric cells, OVA-expressing melanoma-bearing mice showed a decrease in tumor growth and increase in antigen-specific CD8+ T cell infiltration in tumors. with higher median survival compared to other treatments, correlated with activation of iNKT cell and amplification of cancer-specific adaptive immune responses without induction of anergy in iNKTcells. They showed increased CD+4 T cell proliferation in the spleen in response to Exo(aGC-OVA), which was NTK cell-dependent, and activation of the proliferation of Tfh-cells, a higher number of total proliferating germinal center B cells, and plasma cells [92]. The presented study showed promising evidence that the codelivery of diverse antigens may increase the immunogenicity of dexosomes to improve anti-tumor responses to cancer vaccines. The group of Damo et al. showed another approach was to improve the immune-stimulatory properties of dexosomes. They used toll-like receptor (TLR) ligands as adjuvants for maturation of DCs to activate danger signal-sensing pathways to gain the secretion of DEX with a higher capability to activate cytotoxic NK-cells and CD8+ T lymphocytes, which was combined with co-culture of DCs with oxidized necrotic B16F10 cells as the source of tumor antigens. Dexosomes released by such prepared DCs (Dexo(B16 + pIC)) have shown higher potential to induce strong activation of melanoma-specific CD8+ T cells and recruit cytotoxic CD8+Tcells, NK, and NK-T cells to the tumor site. Immunization with Dexo(B16 + pIC) tumor-bearing mice resulted in reduced tumor growth and better survival compared to dexosome-based vaccine formulation, similarly to that tested in clinical trials [93]. These studies show a novel capacity of exosomes as adjuvant carriers, and pave the way for developing new cancer vaccines with boosted immunogenicity and immune response. Recently, Phung et al. designed bifunctional exosomes, putting together two immunotherapeutic vaccines and a checkpoint blockade to improve the anticancer effect. Researchers prepared exosomes from OVA-pulsed DCs matured with poly(I:C) and modified them with anti-CTLA-4 antibodies through the lipid-anchoring method (EXO-OVA-mAb). They examined whether EXO-OVA-mAb can activate T cells in vitro, checked trafficking to lymph nodes in vivo, and its immunostimulatory activity in vivo. Researchers demonstrated that EXO-OVA-mAb induced notable activation of both CD4+ and CD8+ T cells with higher efficiency than other formulations in vitro, which correlated with the highest levels of TNF-a and INFg secreted by EXO-OVA-mAb treated T cells. Moreover, bifunctional exosomes enhanced the proliferation index of both T cell subsets. They showed that EXO-OVA-mAb migrated to the lymph nodes and were efficiently directed to T cells. They also confirmed that administration of EXO-OVA-mAb into mice bearing B16-OVA tumors resulted in robust T cell activation induction. Scientists demonstrated that immunization with EXO-OVA-mAb resulted in the strong Th1 and CTL responses, which notably augmented CD4+TNF-α+ and CD8+IFN-γ+ fractions showed. They also noticed a significantly increased fraction of effector memory T cells (T_EM_ cells) in mice cured with exosomes loaded with OVA and CTL70A+4Ab, compared to other treatments applied in this study. The presented results showed that EXO-OVA-mAb could induce immune responses against OVA-positive tumors. The responses were synergistically enhanced by functionalizing EXO-OVA with CTLA+4Ab, which blocked the immunosuppressive component that inhibits T cell activation. The EXO-OVA-mAb treatment analysis showed increased infiltration into the tumor and enhanced activation of T cells, higher CTLs/Treg ratio, and significant slow-down in tumor progression compared with other studied exosomes [94]. This study demonstrated that combining immunization strategy with checkpoint inhibition is possible and effective. This new approach requires further investigation to determine if this therapy affects survival and inhibits tumor growth.

**Table 2 molecules-27-01303-t002:** Exosome-based cancer vaccines.

EVs Type	EVs Source	Cancer Type	Administration Route	Method	Outcome	References
MAGE-tumor antigens pulsed DEX	autologous DCs	advanced melanoma advanced NSCLC	s.c./i.d.	I phase clinical trial I phase clinical trial	No CD4+, CD8+ T cell responses detected, MAGE-specific T cell response in only ⅓ of patients, increased NK lytic activity in ⅔ of patients	[90,91]
Dexo(B16 + pIC)	TLRs ligands maturated DCs co-cultured with oxidized necrotic B16F10 cells	melanoma	i.d./i.v.	in vivo, mouse model	Reduced tumor growth, prolonged survival, activation of tumor-specific CD8+ T cells, recruitment of CD8+T cells, NK, and NK-T cells to the tumor site	[93]
EXO-OVA-mAb	OVA-pulsed DCs matured with poly(I:C)	melanoma	s.c.	in vitro, and in vivo, a mouse model	Activation and increased proliferation of CD4+ and CD8+ T cells in vitro; strong Th1 and CTL responses are shown by increased CD4+TNF-α+ and CD8+IFN-γ+ fractions and increased T_EM_ cells. higher CTLs/Treg ratio, slowed down tumor progression	[95]

DCs—dendric cells; NSCLC—non-small cell lung cancer; BM-DCs—bone marrow-derived dendric cells; Exo(αGC-OVA)—α-galactosylceramide (αGC)/antigen ovalbumin (OVA)—loaded exosomes; Dexo(B16 + pIC)—exosomes derived from DCs maturated with pIC(TLR3 ligand) and co-cultured with oxidized necrotic B16F10 melanoma cells; EXO-OVA-mAb—exosomes derived from ovalbumin-pulsed dendric cells matured with poly(I:C) (dsRNA analog) and modified with anti-CTLA-4 antibody; B16F10 cells—melanoma cell line; i.d.—intradermal; i.v.—intravenous; s.c.—subcutaneous.

The reviewed studies demonstrated the need and possibility of developing novel anticancer therapies by combining different immunotherapeutic methods and presented exosome-based platforms as a promising strategy for modern cancer vaccine design. There is accumulating evidence that effective cancer immunotherapy must activate the immune system in multiple manners to obtain a solid and long-lasting immune response.

## 6. EVs and Drug Delivery to the Central Nervous System

The brain is protected by a highly selective semipermeable physical border, which maintains tight control over the passage of substances moving from the blood to the central nervous system (CNS). The blood–brain barrier (BBB) only allows the passage of chemical compounds, which are essential to maintain the proper functioning of the nervous system and protect against external factors, which also include various types of therapeutic agents [95]. One of the biggest challenges in neuroscience includes, among others, overcoming the BBB to obtain better treatment results in neoplastic and neurodegenerative diseases. Even the smallest diameter in a drug molecule does not guarantee its beneficial effects. Referring to the comprehensive medicinal chemistry (CMC) database, only 5% of all drugs give the expected results in treating CNS diseases [95,96].

One of the strategies is to influence the BBB and increase its permeability by enhancing paracellular and intracellular transport. The first of them is based mainly on a change in the physicochemical properties of the cell membrane and the extracellular environment, which temporarily allows the opening of the BBB [97]. Unfortunately, implementing these research results in clinical practice was often unsuccessful—the therapeutic effect was not significant, or the BBB disruption caused entrance to the CNS of substances that negatively affected brain function, e.g., cerebral edema and seizures [98,99,100]. Intracellular transportation is mainly based on a temporary change in the chemical properties of the administered chemotherapeutic agent so that it can pass through the endothelial layer. Usually, it is modified to overcome the lipid bilayer, e.g., by creating an inactive, hydrophobic form, or to become a ligand of RTM receptors and pass through the BBB cells toward the brain parenchyma [101,102]. Researchers have been looking for the least invasive drug-delivery method in recent years. As the brain and nervous system are very delicate, the target and amount of the used drugs must be precisely defined. A novel approach was to use carriers filled with the active compound, which would cross the BBB, work at the target site, and not accumulate. Particularly noteworthy were the agents that can carry chemical compounds, such as proteins or lipids and nucleic acids, RNA (mRNA, microRNA), etc., attaining long-term effects. An example of such a solution is the use of nanoparticle carriers and cell-based drug delivery [103,104]. Nanoparticles (NPs) are colloidal, synthetic particles that mainly enter by receptor-mediated endocytosis in the central nervous system [105]. They are characterized by high drug-loading capacity and a controlled release profile [106]. The cell-mediated delivery systems are a suitable option when precise drug administration is needed, especially in the case of tumors [107]. As both proposals, NPs and cell-based drug delivery, have many advantages and have been widely used in treatment, it should be borne in mind that there is significant evidence of their toxicity, high immunogenicity, and low efficiency [23,103,108,109]. Extracellular vesicles (EVs) combine the advantages of both of these delivery systems. As natural carriers, they are integral to numerous biological processes and represent the characteristics of the cells from which they are derived [110,111,112]. This part of the article, devoted to neurological diseases, such as Alzheimer’s and Parkinson’s disease, presents the latest reports on treatment using EVs.

### 6.1. Alzheimer’s Disease

Alzheimer’s disease (AD) is the most common neurodegenerative disease globally. The primary symptom is chronic and incurable dementia associated with loss of memory and independence [113]. AD is caused by an accumulation of the protein fragment β-amyloid outside neurons and twisted strands of the protein tau inside them, leading to neuron death [114]. The complexity of the problem of treating the disease is derived from the effectiveness and safety of the use of drugs that must cross the blood-brain barrier, which is why researchers began to be interested in extracellular vesicles (EVs) as carriers of therapeutic agents. Key aspects in the treatment of AD are the inhibition of amyloid deposition and reduction in brain cell death. Alvarez-Erviti et al. used exosomes to deliver siRNA to the brain to knock down the mRNA and protein BACE1, the major secretase for the generation of β-amyloid peptides. The exosomes were obtained from self-derived dendritic cells to reduce the immune response. The analysis of cortical tissue samples confirmed a significant (over 45%) protein knockdown in a mouse model [56,115]. In another study, the researchers extracted vesicles from human umbilical-cord mesenchymal stem cells (hUC-MSCs) with CD9, CD63, and CD81-positive. In treatment, these exosomes express all of the advantages of hucMSCs, but with lover immunogenicity. The injection of therapeutic vesicles reduced β-amyloid deposition from about 0.6% (in the control group) to 0.2% in the cortex, and from about 0.75% to 0.3% in the hippocampus, due to increased levels of amyloid-degrading enzymes [116]. The MSC-exosomes have also been used for the delivery of miR-223. Wei et al. proved that the therapy reduces cell apoptosis by targeting the PTEN-PI3K/Akt pathway [117]. Yuyama et al. demonstrated how to obtain exosomes from murine neuroblastoma Neuro2a (N2a) cells, and how are they bind to β-amyloid to allow this peptide to be phagocytized in microglia [118]. The neuronal-derived exosomes also proved to be helpful in this peptide decomposition [119]. Therefore, exosomes in their simplicity can be beneficial in Alzheimer’s disease treatment due to their ability to reduce the number of harmful protein deposits and protect neurons.

### 6.2. Parkinson’s Disease

Parkinson’s disease (PD) is one of the most characteristic neurodegenerative diseases. Its most prominent symptom, resting tremor, makes life difficult for more than 9 million people worldwide [120,121]. The pathological cause of PD is mitochondrial dysfunction and excitotoxicity with the lack of oxidoreductase, catalase, and superoxide dismutase [122,123]. The idea of using antioxidants to protect neurons by inhibiting the inflammatory response was realized using catalase-loaded exosomes (exoCAT). Haney et al. confirmed that exoCAT demonstrated neuroprotective activity—they can eliminate reactive oxygen species (ROS) produced by activated macrophages. The experiment also showed that the exoCAT obtained by sonication has a more significant beneficial effect than exosomes loaded with catalase by freeze/thaw cycles [123]. Symptoms of Parkinson’s disease also stem from dopamine deficiency. The intravenous administration of free dopamine in minimal effective concentration is highly toxic for the human body. The enclosure of the compound in the exosomes caused significantly lower toxicity than free dopamine with greater concentration in the brain [124]. In the samples of brain tissue with PD, characteristic histopathological features can be observed, e.g., α-synuclein toxic aggregates are known as a part of Lewy bodies [125]. Cooper et al. designed three siRNA sequences to downregulate endogenous α-Syn peptide and mRNA levels in the brain. Exosomes delivered the siRNA via injection [126]. A similar procedure was performed using shRNA-MC as therapeutic cargo in another experiment. The collected data confirmed decreased alpha-synuclein aggregation and neuron loss [127]. The results of these and previous studies confirm the therapeutic potential of using exosomes for the long-term treatment of neurodegenerative diseases.

### 6.3. Stroke

Stroke, a cerebrovascular incident manifested by sudden onset of a focal neurologic deficit, is the most disabling chronic condition. A healthy lifestyle could prevent many new stroke cases through everyday physical activity and the care of a GP. The most significant influence on the progression of the disease is broadly understood to be oxidative stress, which also contributes to many other disorders that accompany strokes, such as hypertension, atherosclerosis, or diabetes. After ischemia, many proinflammatory and cytotoxic interleukins and factors can cause cell damage or even death. Furthermore, in this case, researchers found the use of vesicles for therapeutic purposes. Liu et al. indicated that the simultaneous use of exosomes with transferrin and enkephalin accelerated the overcoming of the blood–brain barrier and promoted neuron regeneration. This combination decreased the levels of LDH, p53, and caspase-3, master regulators of apoptosis. In addition, results obtained in vitro showed that tar-exo could passively enter neurons [128]. These cells are essential for functional recovery after stroke, and therefore, it is crucial to support their reconstruction. For this purpose, the influence of miR-133b on their rebuild was investigated. Obtained data suggest that the increased exosome miR-133b from MSCs promotes neurite outgrowth by regulating cell proliferation and apoptosis [129]. Similar effects were noted with the miR-17-92 cluster in exosomes—it modulates the PI3K/AKT/mTOR/GSK-3β signaling pathway [130]. In addition, Otero-Ortega et al. proved that therapy with exosomes promotes the differentiation of oligodendrocyte progenitor cells. The applied treatment increased all white matter repair-associated markers (CNP-ase, A2B5, and MOG) levels. The exosomes harvested from MSCs also contributed to the restoration of myelin sheaths, which confirms that the therapy supports the formation of nerve connections at every stage [131]. To restore the physiological functions of the central nervous system after ischemia-reperfusion injury, the well-known compound curcumin was used. This chemical compound was encapsulated in mouse embryonic stem-cell exosomes (MESC-exo) and applied by intranasal administration. The collected data confirmed that MESC-exo^cur^ treatment reduced inflammation by decreasing levels of ROS, TNF-α, and malondialdehyde (MDA), and increasing levels of glutathione (GSH) [132].

### 6.4. Traumatic Brain Injury (TBI)

Traumatic Brain Injury (TBI) is a disruption in the brain’s normal function that is usually caused by mechanical damage. There are many types of consequences, e.g., hematoma, contusion, intracerebral and subarachnoid hemorrhage, and ischemia. Often the lesions are irreversible and cause permanent damage. Unfortunately, an effective treatment has not yet been found. In this condition, the most important factor is to protect the existing neurons and support the reconstruction of brain tissue. One of the promising solutions appears to be the supply of agents that reduce local inflammation, and consequently, promote better blood and cerebrospinal fluid circulation. Williams et al. confirmed that exosomes derived from a human mesenchymal stem cell (MSC) show this effect with early single-dose treatment. The research was carried out on an animal model. Laboratory tests confirmed a decrease in the levels of inflammatory markers (IL-1, IL-6, IL-8, and IL-18). Additionally, it was observed that the therapy significantly reduced intracranial pressure (23 ± 1.5 mm Hg in a control group; 16 ± 1.7 mm Hg in a tested group) [133]. It is also worth noting that these clinical trials were conducted 7 days after applying the therapy. A behavioral study also investigated MSC-derived exosomes and their effects on brain function. It has been proven that exosome administration significantly enhances spatial learning, the impairment of which is hard evidence of hippocampal injury [130]. The same study confirmed that modified EVs increase vascular density, angiogenesis, and neurogenesis. The results of these studies are optimistic and give hope to many people affected by TBI.

These examples show how many possibilities are offered by using EVs in therapy against diseases of the Central Nervous System. It should be emphasized that loading EVs with various active molecules is not the only way to obtain specific treatment effects. It is often enough to bring EVs from particular cells, thanks to which they have valuable properties. This simplicity may be the key to developing therapies with the safe use of autogenous and non-immunogenic Evs. Often, the symptoms are caused by the disease itself and by the human body reaction and inflammation, which is not desired, especially in the central nervous system. The table below presents therapies using both EVs with additional cargo and unmodified EVs (Table 3).

## 7. Application of EVs in Dermatology and Medical Aesthetics

Skin is the largest organ of the human organism (the total area is 1.5–2 m^2^), and is composed of three main layers, i.e., epidermis, dermis, and hypodermis [137]. It provides a barrier between our internal molecular mechanisms and the external environment, including physical, chemical, and biological harmful external factors. Furthermore, it participates in essential processes, such as system thermoregulation, regulation of water-mineral balance, and perception of external stimulus (e.g., touch, pain, heat, cold) [138,139,140]. It is also a significant element of the innate immune system, creating a barrier that isolates the body against pathogenic microorganisms. Skin diseases can be severe problems, ranging from minor cuts to diseases, such as melanoma, skin burns, and chronic ulcerations. These situations significantly affect the physical aspects of the human body and patients’ psychological comfort. Therefore, therapeutic solutions should be sought to quickly and effectively cure the patients, or significantly reduce symptoms.

Extracellular vesicles (EVs), ubiquitous in the human organism and secreted by almost all cells, are gaining increasing scientific interest in many fields, including dermatology and aesthetic medicine. Their bioactive cargos are transferred, mediating in cell–cell communication. Thereby, EVs are involved in many cellular processes, e.g., cell proliferation and differentiation, cell death, angiogenesis, and immune regulation [11]. They play a significant role in the pathogenesis of a wide range of inflammatory skin disorders [11,141,142,143,144,145].

Nonetheless, because of their intrinsic ability to cross cellular and tissue barriers (e.g., blood-brain barrier), EVs are remarkably effective drug-delivery systems. Shao et al. observe that in addition to their native bioactivity, EVs can be engineered to deliver a wide range of different proteins, nucleic acids, and/or chemicals or drugs [146,147]. Moreover, their surface can also be re-engineered, thus allowing high target specificity [148]. It is predicted that their utilization as biomarkers and cell-free therapeutic drugs will increase significantly over the next decade. In accordance with the fact that most EV functions are associated with the immune response, their role as anti-inflammatory therapeutics for inflammatory skin diseases (including chronic wound-healing, psoriasis, and atopic dermatitis (AD)) is emerging.

### 7.1. Wound Healing

Cutaneous wound healing is an elaborate process aimed at rebuilding and restoring the functions of the damaged tissues. Its course can be divided into four predictable stages: (1) blood clotting (hemostasis), (2) inflammation, (3) tissue growth (cell proliferation and migration) associated with angiogenesis, and (4) remodeling/maturation [149]. The foundation of the proper course of the healing process is the correct sequence and duration of the phases mentioned above. For instance, disorders in the inflammation phase may result in chronic skin ulcers (CSU), and increased fibrosis can give rise to hypertrophic scarring that may develop into keloids [150,151]. Furthermore, the correct process of angiogenesis responsible for delivering oxygen and nutrition to the wound site allowing the proliferation of fibroblasts, production of collagen, and re-epithelialization, is essential for wound healing [152]. The number of CSUs (also known as chronic cutaneous ulcers (CCU)) is constantly growing in economically developed countries. The two main reasons for this are aging societies and increasing numbers of patients diagnosed with diabetes [153]. It is projected that in 2030, the number of people aged >60 years will reach 1.4 billion (~16.5% of the global population) [154]. Diabetes mellitus (DM) reached epidemic proportions worldwide over the last few years, with 463 million adults (20–79 years) diagnosed [155]. Forecasts report that this number will rise to 700 million by 2045. Nearly 70% of diabetic patients suffer from dermopathy [156]. Over time, many diabetic wounds become CSU/CCU (e.g., foot ulceration) in many cases (~80%) resulting in amputations of limbs [157]. Unfortunately, this phenomenon’s exact molecular mechanism is still not fully understood. However, the key complications include impaired angiogenesis processes, decreased capillary density and vascularity, hypoxia, neuropathy, and damage associated with the reactive oxygen species (ROS) activity [139,158,159]. Due to the negative impact on the patients’ mental health, and the fact that chronic wounds increase the risk of other systemic disease development, there is an urgent need to obtain effective therapeutic solutions.

Over the last decade, the promising potential of exosomes, mainly derived from mesenchymal stem cells (MSCs) for regenerative medicine, has been reported. MSCs have the ability of self-renewal and multipotent differentiation. A wide range of tissues can provide MSCs, from bone marrow and fatty tissue to synovium and periodontal ligaments [139,153]. Additionally, after extraction, they can be cryopreserved without significant loss of their therapeutic potential over time. Unfortunately, no specific biochemical marker for MSC has been defined so far. However, the scientific consensus is that they should not exhibit the expression of hematopoietic and endothelial markers, such as CD11b, CD14, CD-31, CD-33, CD-34, CD-45, and CD-133 [160]. MSCs are considered a promising candidate due to their low immunogenicity and immunomodulatory properties [161].

Nonetheless, it has been hypothesized that the primary source of the regenerative functions of MSCs is not their ability to differentiate into a wide variety of cells, but their paracrine activity [153]. This hypothesis was supported by several studies that showed that the media used for MSCs culture had a similar or even higher regenerative capacity than the MSCs themselves. Interestingly, among the components of the MSCs secretome (including the free fraction made of soluble factors and metabolites), the parts encapsulated in molecular vesicles (MVs), especially in EVs, are the main factors responsible for the therapeutic properties of the conditioned media from MSC cultures [153,162]. Ding et al. proposed a solution based on the exosomes originating from bone-marrow-derived MSCs (BMSCs) preconditioned by deferoxamine (DFO-Exos) [163]. The presented study investigated whether the BMSCs-DFO-Exos exhibited superior proangiogenic property in wound repair in vitro and in vivo. Scientists revealed that DFO-Exos stimulate angiogenesis in vitro by activating the PI3K/AKT signaling pathway via PTEN downregulation mediated by miR-126.

Moreover, this hastened angiogenesis and the entire wound healing process in streptozotocin-induced diabetic rats in vivo. Liu et al. found that exosomes derived from human umbilical-cord mesenchymal stem cells (hucMSC-Exs) accelerated the wound-healing process in a rat model of deep second-degree burn injury [161]. The obtained results suggest that treatment with hucMSC-Exs increased expression of CD31 in vivo and discovered that husMSC-Exs include angiopoietin-2 (Ang-2). The use of hucMSC-Exs for tissue repair results in enhanced expression of the Ang-2 protein in the wound area and human umbilical-vein endothelial cells (HUVECs) through exosomal-mediated Ang-2 transfer.

It should be mentioned that the primary mode of EV delivery is a local or systemic injection of free EVs. Unfortunately, it results in rapid clearance of EVs. One approach to overcoming this limitation, prolonging EVs retention, and enhancing therapeutic efficacy is re-engineering the surface to improve the target specificity. Furthermore, in recent years, increasing attention has been given to EV-loaded biomaterials (e.g., hydrogels) as dressings for wounds, and the outcomes are promising. This approach allows for a hefty dose of vesicles to the target tissue. The development of biocompatible scaffolds that ensure the sustained release of exosomes while maintaining a high level of bioactivity at the target site significantly improves the healing process of chronic wounds. A wide range of natural biomaterials, synthetic polymers, and ceramics have been adapted for EVs loading [159,164,165,166].

### 7.2. Atopic Dermatitis (AD)

Atopic dermatitis (AD), generally known as atopic eczema, is a typical inflammatory skin disorder associated with chronic uncontrolled inflammatory responses [167]. It is prevalent in ~8% of adults and ~20% of children [168]. Unfortunately, the exact molecular mechanism of disease pathogenesis remains unclear. However, pathogenesis includes genetics (e.g., loss-of-function mutations in the *FLG* gene), immune signaling dysfunctions, and difficulties with the permeability of the skin barrier (allowing the entry of allergens, environmental pollutants, and pathogens) [168,169]. The disease mainly appears through severe pruritus and erythematous lesions, which significantly adversely affect the quality of patients’ lives [167]. AD treatment is primarily based on pharmacological intervention, including corticosteroids and calcineurin inhibitors. This kind of solution results in various adverse effects and the development of drug resistance. Therefore, there is an urgent need to develop new, more effective, and less toxic therapies. As with chronic wound healing, MSCs have also found a use in AD healing. The application of these stem cells in treating AD has been completely and exhaustively described by Daltro et al. [167]. However, also for AD therapy, exosomes obtained from MSC are their compelling alternative. They have identical biological functions to the cells from which they are extracted, while having lower immunogenicity and higher stability [139]. This avoids most of the problems associated with live MSC-based therapy.

Over the last few years, increased susceptibility to AD has been chiefly ascribed to elevated levels of Th2 cytokines (inside-outside hypothesis) [139,170,171]. Accordingly, most of the research has focused on reducing Th2-mediated immune responses. Nevertheless, recent studies have shown a strong correlation between dysregulation of immune response (e.g., elevated Th2 cytokine levels) and defective epidermal barrier caused by abnormal gene expression. Seung Cho et al. demonstrated that human adipose-tissue-derived mesenchymal stem-cell-derived exosomes (ASC-exosomes) could be a novel cell-free therapeutic modality for AD treatment [172]. Using an in vivo mouse model, scientists revealed that ASC-exosomes could ameliorate AD symptoms through, i.e., reducing mRNA expression of various inflammatory cytokines, such as interleukin (IL)-4, IL-23, IL-31 and tumor necrosis factor-α (TNF-α). Furthermore, Shin et al. showed that subcutaneous injection of ASC-exosomes in an oxazolone-induced dermatitis model also considerably minimized the levels of IL-5, IL-13, TNF-α, IFN-γ, IL-17, and TSLP [171].

Another AD therapeutic target is based on impaired protective enzyme-mediated response against increased ROS levels in dermatitis patients. Yang et al. extracted EVs from human umbilical-cord blood-derived MSCs (hUCB-MSCs), introduced with superoxide dismutase 3 (SOD3), one of the superoxide dismutase protein family members. It is an antioxidant enzyme that catalyzes superoxide radicals [173]. The research was conducted on the murine dermatitis model. It was demonstrated that transduction of SOD3 in hUCB-MSCs enhanced cells’ immunoregulatory and therapeutic functions against in vitro immune cell activation and in vivo dermatitis manifestation, respectively. Furthermore, EVs from modified cells delivered SOD3, which efficiently regulated excessive inflammation in vitro and in vivo.

Interestingly, despite their role in pathogenesis and their potential role as a therapeutic target for managing AD aggravation [11,174], bacterial EVs have potential in AD treatment. Kim et al. suggested the preventive potential of *Lactobacillus plantarum*-derived EVs in skin inflammation induced by *Staphylococcus aureus*-derived EVs [175].

### 7.3. Psoriasis

Psoriasis is the most common chronic inflammatory skin disease, and affects nearly 125 million people worldwide [176]. The disorder is associated with impaired proliferation and differentiation of keratinocytes. Additionally, massive infiltration of the immune system cells has been noted [11]. It appears through red squamous plaques localized on the elbows, knees, sacroiliac region, nails, and scalp [176]. Unfortunately, the pathogenesis of psoriasis remains unclear, but previous research reports that the disease is mediated by T cells and dendritic cells. Its immunology has been comprehensively described by Lowes et al. [177]. As in AD, EVs are also involved in the pathogenesis of psoriasis, as has been extensively characterized by Shao et al. [11]. The utilization of EVs as therapeutic agents for psoriasis management is still under investigation. However, Zhang et al. described promising outcomes for human umbilical cord mesenchymal stem-cell-derived exosomes (hucMSCs-Exo) [178]. Scientists have revealed the ameliorating effect of hucMSCs-Exo on imiquimod (IMQ)-induced mice. Subcutaneous hucMSCs-Exo injection reduced the level of STAT3/p-STAT3, IL-17, IL-23, and CCL20. Furthermore, exosomes suppressed dendric cell (DCs) maturation activation and blocked the positive feedback effect of IL-17 on keratinocytes. Moreover, inhibition of IL-23 secretion was also noted.

### 7.4. Cutaneous Medical Aesthetics

Nowadays, people pay more attention to outward appearance. Therefore, hair loss, scars, or visible skin aging processes often lower people’s self-esteem. This has fueled the development of aesthetic medicine, which constantly seeks new, less invasive, and at the same time effective methods of cosmetic intervention. Promising outcomes with exosomes in regenerative medicine have resulted in increased investigation of EV utilization in skin rejuvenation, scar removal, and hair loss.

Distortions during different phases of wound healing can lead to the recruitment and excessive activation of myofibroblasts [179]. This may result in the formation of hypertrophic or keloid scars. Thus, the prevention of scar formation is a challenge for many patients. Adipose mesenchymal stem cells (ASCs) play a crucial role in the process of tissue regeneration. Recent studies have shown that exosomes extracted from ASCs (ASC-Exos) might be a novel method for scarless wound repair. Wang et al. demonstrated that intravenous injection of ASC-Exos enhanced extracellular matrix (ECM) reconstruction through regulating the ratios of collagen type III: type I, TGF-β3:TGF-β1, and MMP3:TIMP1 in a mouse skin incision model [180]. Zhu et al. presented the use of human adipose-derived stem cells (hASCs) as a new approach to preclude hypertrophic scar development [181]. Local hASCc injection successfully prevented hypertrophic scar formation by suppressing myofibroblast aggregation and collagen deposition in a rabbit scar model.

Interestingly, the histological analysis by Hu et al. revealed that ASC-Exos-administration might improve collagen I and III production in the early stages of wound healing [182]. Simultaneously, the same ASC-Exos might decrease collagen expression to reduce scar formation. Additionally, MSC-derived EVs exhibit the ability to prevent scars [183,184,185]. Shojaati et al. presented outcomes regarding EVs extracted from corneal stromal stem cells (CSSC) and mesenchymal stem cells [186]. Treatment resulted in a decrease in the expression of fibrotic genes *Col3a1* and *Acta2*, blocked neutrophil infiltration, and restored normal tissue morphology. The results supported a hypothesis that EVs appear to serve as a delivery vehicle for miRNA, which affects the regenerative action.

EVs are used in a wide range of medicine, including anti-aging skin rejuvenation. Yoshida et al. demonstrated an extremely intriguing study [187]. Scientists revealed that EVs extracted from young mice contain significantly higher extracellular nicotinamide phosphoribosyltransferase (eNAMPT) levels than older ones. eNAMPT is an enzyme playing a crucial role in nicotinamide adenine dinucleotide (NAD+) biosynthesis, which is involved in anti-aging processes. It has been shown that the level of circulating eNAMPT also declines with age in humans. Yoshida et al. showed that the intracellular NAD+ levels in primary hypothalamic neurons increased after treatment with EVs from the plasma of young mice, in contrast to older mice. Moreover, the intraperitoneal administration of plasma EVs from young mice to old mice resulted in a significant improvement in their physical activity (wheel-running activity) and an increase in life expectancy by about 10%.

Macrophage EV (MAC-EV) treatment revealed the potential of the EV as a novel anagen inducer in hair loss treatment [188]. MAC-EVs improved proliferation and migration and resulted in a higher level of hair-inductive markers of dermal papilla (DP) cells. DP cells treated with MAC-EVs showed elevated expression of vascular endothelial growth factor (VEGF) and keratinocyte growth factor (KGF). Furthermore, MAC-EVs promoted hair follicle (HF) growth in vivo.

The studies described above show that EVs are currently being extensively investigated in a wide range of medicine, including dermatology and aesthetic medicine. The presented research shows the tremendous potential and wide range of applications of EVs. Their diagnostic and therapeutic purposes in atopic dermatitis and wound-healing give promising results and bring this area of dermatology to the next level of development, allowing the use of these cell-free therapies. Moreover, studies into their use in the treatment of other diseases in this area are also rapidly developing. However, many molecular mechanisms underlying these phenomena remain unclear and require further research. The table below summarizes the therapies used to treat various skin diseases presented in this work (Table 4).

## 8. Possible Limitations of the Therapy

Therapies with extracellular vesicles (EVs) are gaining more and more popularity and a group of supporters. The vast majority of researchers, comparing treatments using stem cells and EVs derived from these cells, distinguished many advantages of EVs, including their lower immunogenicity and toxicity, which is of particular importance in treating autoimmune diseases [189].These features depend on the origins of EVs and their content. They are already known at the design stage of the entire therapy process, because the vesicles ‘inherit’ these types of properties from the cell from which they were isolated. At this point, a decision is also made as to whether the EVs used will be auto- or allogeneic in the case of experimental therapies, depending on the type of disease and the patient’s condition [190]. This also allows estimation of the risk of an immune reaction, which always exists.

The process of their isolation also has a significant impact on EVs, and is expensive and time-consuming, shows how much valuable material is lost in its subsequent stages, and affects the therapeutic properties. EVs may fail during the preparation, but may also gain features that may be potentially dangerous for the patient and cannot always be predicted at the design stage. For example, Beckerman’s team proved the importance of the centrifuging stage. It emerged that this process affects phosphatidylserine, present in abundance in the vesicle membranes, which increases the likelihood of blood clotting and thus the formation of dangerous clots [191,192].Studies have also been conducted in this direction, confirming that heparin did not abolish the pro-coagulatory action of EVs [193]. The environment of vesicle release is also of great importance, as it differs between experiments with cell culture and animal studies or preclinical studies. It affects the quality of the EVs and the amount released, which may cause problems in obtaining the target amount of microbubbles for use in experimental therapies [189,194].

Other difficulties, apart from selecting the cells from which EVs will be derived and the isolation process itself, may be encountered when choosing the technique of EV administration (Figure 1). Intravenous injection makes them more accessible to the immune system and phagocytoses by macrophages or Kupffer cells [192,195]. EVs can also accumulate in various organs, e.g., they can cross the blood-brain barrier [a]. The place where the vesicles accumulate also depends on their quality and origin [196]. In many cases, this is a therapeutic effect, however, it must be taken into account that it may also cause side effects, such as local inflammation [197]. To avoid the accumulation of vesicles in the internal organs, the team of Gomzikova et al. recommends administering them intramuscularly or subcutaneously in justified cases. According to the obtained data, intravenous administration of EVs leads to the formation of deposits in organs within 48 h, while intramuscular and subcutaneous injections extend this time to 14 days [198].

The above examples perfectly illustrate the diversity of the EV group, and how much their diversity depends on the characteristics of the cells from which they come [190]. As carriers of natural origin, they are sensitive to all kinds of changes in the environment, including changes in pH and oxygen content [199]. A significant challenge is determining the degree of safety of using Evs as a therapeutic agent in in vitro studies. Due to the various individual characteristics that each of us has, carefully planned and long-term clinical tests are necessary to confirm the designed therapies’ effectiveness and legitimacy.

## 9. Conclusions and Future Perspective

Studies of extracellular vesicles (EVs) as potential therapeutic tools for difficult-to-treat disorders are rapidly growing. EVs are applicable in many fields of medical and clinical sciences, such as dermatology and neurology. Due to their versatility and the possibility of the oncology mentioned in the above modification, they are often used as carriers of potentially therapeutic drugs, especially those that find it challenging to obtain a therapeutic concentration at the site of action, or to overcome various barriers in the human body. As the research was carried out, it began to be noticed that more benefits are brought not by multiple drugs and chemical compounds, but by structures such as RNA (mainly miRNA and siRNA), various inhibitors, and cytokines. These factors make it possible to obtain a long-lasting healing effect, because they affect the symptoms of the disease and often treat the underlying cause.

EVs appear as perfect candidates to deliver therapeutic drugs, presenting several advantages compared with liposomes or polymer-based techniques. Firstly, they usually exist in body fluids, and consequently, they are stable in physiological conditions. Furthermore, they are less immunogenic and cytotoxic compared to polymerized vectors. Finally, EVs can transfer cargo to particular cells due to their membrane proteins and lipids that can apply them to specific receptors in the target cells. In particular, this aspect is essential to transport therapeutics to particular areas where polymeric vesicles or liposomes are unable to reach, such as the brain. In addition, some studies indicate that exosomes may cross the BBB following active endocytosis mechanisms. Moreover, EVs can interact with the BBB, changing the barrier’s characteristics [167].

EVs have vast therapeutic potential, considering their low immunogenicity and abounding protein or gene transfer capacity. Recently, new strategies to produce ad hoc exosomes have been devised. Cells delivering exosomes have been genetically engineered to overexpress particular macromolecules, or transformed to release exosomes with appropriate targeting molecules. In one strategy, cells from the patient would be placed in culture and genetically modified as required for targeting therapeutic protein/RNA. In the next step, EVs would be isolated from cultures and administered to the patient. In a second strategy, the genetically modified cells in vivo produce therapeutic EVs [170].

Nevertheless, there are significant difficulties to solve during EV application as drug-delivery agents in the clinic. Firstly, a considerable challenge is the low isolation yield obtained and the presence of some impurities, such as proteins or other EVs. Secondly, it is crucial to analyze the influence of EVs in physiological and pathological conditions to predict short- and long-term safety [171]. In the case of anti-cancer therapies, it is necessary to differentiate between tumor-cell- and healthy-cell-derived vesicles. Innovative technologies, such as microarrays, specific monoclonal antibodies, and RNA markers amplification strategies, have already been set up for these purposes [172]. An essential point in developing EV-based carriers is their stability in body fluids. It was observed that the EVs were stable for at least three months at 37 °C, 4 °C, −20 °C, and −80 °C [171].

EVs consist of various (closely related and often nanoparticulate) structures that cannot be isolated and fully quantified. It is also impossible to indicate precisely which structural elements determine their therapeutic effectiveness. Their composition, quality, and operation in vivo are mainly dependent on the processes of their production, which are not repeatable, as in the case of chemical process nanocarrier synthesis. In this respect, EVs have similar features to NBCDs (non-biological complex drugs). NBCDs treat many disorders, including cancer, autoimmune diseases, infectious diseases, anemia, and more. NBCDs are a rapidly evolving field of innovative drugs with advanced site- and rate-specific release properties for targeted delivery and targeted formulations [171].

Moreover, industrial-scale manufacture of therapeutic exosomes is now possible. BIA Separations uses solutions for the extraction of exosomes on an industrial scale. Such clinical applications require obtaining thoroughly purified exosomes and reducing the amount of non-exosomal vesicles to a trace level. The new BIA Separations technology provides new tools for this purification while ensuring fast and efficient process development. The technology can be scaled from the lab to production. Cornerstone Exosome Development Solution is based on the possibilities of two innovative technologies. The first is Kryptonase™ which is an integrated enzymatic treatment that reduces the DNA of the host cells and facilitates the removal of protein contaminants from the host cells. The second technology uses a CIMmultus™ chromatographic column that eliminates non-exosomal vesicles and concentrates exosomes in a low-shear environment.

## Figures and Tables

**Figure 1 molecules-27-01303-f001:**
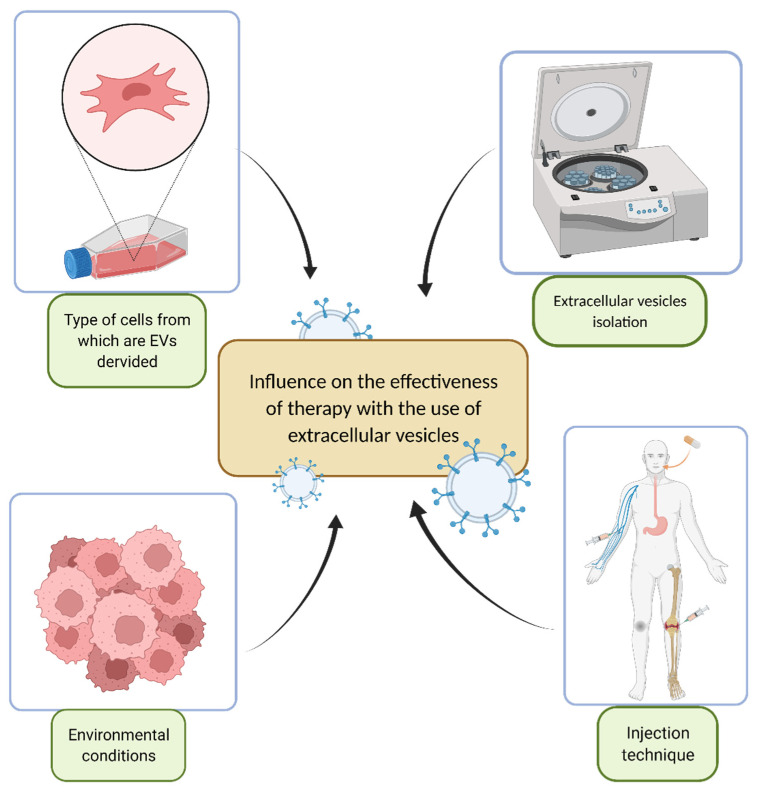
Factors that influence the effectiveness of treatment with extracellular vesicles. The origin of EVs affects their features, such as their immunity, toxicity, and therapeutic properties, as they bear the hallmark of the parental cells. Isolation procedures, such as centrifugation techniques, may cause changes in EV membrane structure that lead to the loss and gain of features that may be dangerous for the patient and cannot always be predicted at the design stage. There are differences in experimental conditions for vesicle release between cell culture studies, animal studies, and preclinical studies. The environmental conditions of EV release affect the quality and quantity of EVs; therefore, experimental therapies may have difficulty achieving the target amount of EVs. The injection technique of EV administration affects their pharmacokinetics and bio-distribution. Intravenous injection makes EVs more accessible to the immune system and prone to phagocytosis. The accumulation of EVs in the organs is higher after intravenous injection than after intramuscular or subcutaneous injection, which affects the therapeutic response.

**Table 1 molecules-27-01303-t001:** EVs in cancer therapy.

Cancer Type	Source of EVs	Cargo	Administration Route	Method	Outcome	References
PDAC	fibroblast-like mesenchymal cells	siRNA/shRNA specific to oncogenic KrasG12D	i.p.	in vivo mouse model	Inhibition of tumor growth, prolonged lifetime in advanced stage	[78]
Pancreatic cancer	NK cells	miR-3607-3p	N/A	in vitro	Inhibited cell viability, proliferation and migration	[79]
Pancreatic cancer	BM-MSCs	Galectin-9 siRNA, OXA	i.v. (tail vein injection)	in vivo mouse model	Inhibition of tumor growth, reduction in tumor size, increased level of CD8+cytotoxic T cells population, decreased ratio of T-reg cell	[80]
Pancreatic cancer	BM-MSCs	miR-124	i.v.	in vivo mouse model	Inhibition of tumor growth	[86]
Breast cancer	MSCs	Taxol	i.v.	in vivo mouse model	Inhibition of tumor growth, inhibition of metastases	[82]
ERBB2-positive breast cancer	Expi293 cells	Surface anti-CD3-anti-HER2 scFv antibody	i.v.	in vitro, in vivo mouse model	Activation of T-lymphocytes Inhibition of tumor growth	[83]
Breast cancer	LPS stimulated CD90^low^ADSCs	miRNA-16-5p	i.p.	in vivo mouse model	Inhibition of cancer growth	[84]
Metastatic TNBC	autologous breast cancer cells	SiRNA (CBSA/siS100A4)	i.v.	in vivo mouse model	Suppression of postoperative metastasis	[85]

BM-MSCs—bone marrow-derived mesenchymal stem cells; MSCs—mesenchymal stem cells; ADSCs—adipose-derived mesenchymal stem cells; Expi293 human kidney cells derived from the 293 cell line; CBSA—cationic bovine serum albumin, TNBC—Triple-negative breast cancer; PDAC—pancreatic adenocarcinoma; i.d.—intradermal; i.p.—intraperitoneal; i.v.—intravenous; s.c.—subcutaneous.

**Table 3 molecules-27-01303-t003:** EVs in neurological injuries.

Diseases of the Nervous System	Source of EVs	Cargo	Administration Route	Method	Outcome	References
Alzheimer’s disease	Self-derived dendritic cells	GAPDH siRNA, BACE1 siRNA	i.p.	in vivo mouse model	Inhibition of amyloid creation and reduction in brain-cell death	[56]
Alzheimer’s disease	hUC-MSCs	N/A	N/A	in vitro	Inhibition of amyloid deposition Promote the Secretion of Aβdegrading enzymes Reduction in brain cell death	[116]
Alzheimer’s disease	hUC-MSCs	miR-223	N/A	in vitro	Inhibition of apoptosis of neurons in vitro by targeting PTEN	[117]
Alzheimer’s disease	Murine neuroblastoma Neuro2a (N2a) cells	N/A	i.c.	in vivo mouse model	Reduction in Aβ levels Inhibition of amyloid deposition Reduction in the Aβ-mediated synaptotoxicity	[123]
Parkinson’s disease	RAW 264.7 macrophages	Catalase	i.c.	in vivo mouse model	Neuroprotective activity Inhibition of oxidative stress	[123]
Parkinson’s disease	Blood	Dopamine	i.v.	in vivo mouse model	Reduction in dopamine toxicity	[124]
Parkinson’s disease	Murine dendritic cells	α-Syn siRNA	i.v.	in vivo mouse model	Reduction in α-Syn protein aggregates	[126]
Stroke	BMSC	Enkephalin	i.v.	in vivo rat model	Inhibition of neuronal p53/Caspase-3 Decreased levels of LDH Promotion of neuroregeneration	[128]
Stroke	MSC	miR-133b	i.v.	in vivo rat model	Promotion of neurite outgrowth	[134]
Stroke	MSC	N/A	i.v.	in vivo rat model	Treatment prevents the post-stroke brain damage (mNSS test)	[135]
Traumatic Brain Injury	MSC	N/A	i.v.	in vivo swine	Reduction in the levels of inflammatory markers (IL-1, IL-6, IL-8, and IL-18) Reduction in intracranial pressure	[133]
Traumatic Brain Injury	MSC	N/A	i.v.	in vivo mouse model	Reduction in the level of inflammatory marker IL-1β	[136]

hUC-MSCs—human umbilical cord mesenchymal stem cells; GAPDH—glyceraldehyde 3-phosphate dehydrogenase; BMSC—Bone Mesenchymal Stem Cells; LDH—Lactate dehydrogenase. i.v.—intravenous; i.c.—intracranial; s.c.—subcutaneous.

**Table 4 molecules-27-01303-t004:** Selected EVs in dermatology and aesthetic medicine.

Disease	Source of EVs	Cargo	Administration Route	Method	Outcome	References
Diabetic Wounds Chronic Skin Ulcers (CSU) Chronic Cutaneous Ulcers (CCU)	BMSCs preconditioned by DFO	N/A	s.c.	in vitro (HUVECs) in vivo rat model	Stimulation of angiogenesis, Activation of the PI3K/AKT signaling pathway via miR-126 mediated PTEN downregulation to stimulate angiogenesis in vitro	[163]
Diabetic Full Thickness Cutaneous Wounds	AMSCs mixed with FHE hydrogel (FHE@exo)	N/A	hydrogel dressing	in vitro (HUVECs) in vivo mouse model	Promotion of proliferation, migration, and tube formation ability of HUVECs Decrease in scar formation and enhancement of wound healing efficiency in vivo	[159]
Wound Healing	hUCB-derived plasma	N/A	s.c.	in vivo mouse model in vitro HSF and HMEC	Enhancement of angiogenesis and re-epithelialization Reduction in scar widths Promotion of the proliferation and migration of fibroblasts Enhancement of the angiogenic activities of endothelial cells High miR-21-3p expression in UCB-Exos Inhibition of phosphatase and PTEN and SPRY1 by miR-21-3p	[185]
Deep second-degree skin burns	hucMSCs	Ang-2	s.c.	in vitro (HUVECs) in vivo rat model	Wound-closure rate improvement Increase in CD31 expression in vivo hucMSC-Ex-delivered Ang-2 exerts proangiogenic in cutaneous wound healing	[161]
Atopic Dermatitis	ASCs	N/A	i.v or s.c.	in vivo mouse model	Reduction of the number of infiltrated mast cells and CD86+ and CD206+ cells Reduction inserum IgE Reduction in (IL)-4, IL-23, IL-31 and TNF-α mRNA expression	[171]
Atopic Dermatitis	ASCs	N/A	s.c.	in vivo murine model	Reduction in trans-epidermal water loss Enhancement of SC hydration Decrease in IL-4, IL-5, IL-13, TNF-α, IFN-γ, IL-17, and TSLP levels Induction of the ceramides and dihydroceramide production	[172]
Psoriasis	hucMSCs	N/A	s.c.	in vivo murine model in vitro DC and HaCaT	Decrease in STAT3/p-STAT3, IL-17, IL-23, and CCL20 levels in vivo Suppression of the DC maturation and activation in vitro (DC) Inhibition of the IL-23 Secretion in vitro (DC) Reduction in STAT3/p-STAT3, IL-17, IL-23, and CCL20 levels in vitro (HaCaT)	[181]
	hP-MSCs encapsulated by CS hydrogel	N/A	s.c.	in vivo mouse model	Aging DFLs function amelioration Promoted senescent fibroblasts proliferation process Enhancement of the synthesis of ECM proteins Inhibition of the MMPs overexpression Enhanced collagen expression Decreased expression of SAS-related factors Restoration of tissue structures	[123]
Hair Loss	MACs	N/A	i.d.	in vivo mouse model in vitro DP cells	Activation of Wnt/β-catenin signaling pathways by Wnt proteins presented MAC-EVs, and transcription factors (*Axin2* and *Lef1*) Increase in VEGF and KGF expression levels in DP cells Promotion of HF growth in vivo Improvement in hair shaft size in human HF	[149]

BM-MSCs—bone-marrow-derived mesenchymal stem cells; MSCs—mesenchymal stem cells; AMSC—adipose mesenchymal stem cells; hucMSCs—human umbilical-cord mesenchymal stem cells; hP-MSCs—human placental mesenchymal stem cells; hUCB-MSCs—human umbilical-cord blood-derived mesenchymal stem cells; hUCB—human umbilical-cord blood plasma; HUVEC—human umbilical-vein endothelial cells; HaCaT—human normal keratinocytes; DC—dendritic cells; HSF—human skin fibroblasts; HMEC—human microvascular endothelial cells; DP cells—dermal papilla; MAC—macrophages; DFO—deferoxamine; PTEN—phosphatase and tensin homolog; Ang-2—angiopoietin-2; FHE hydrogel—polypeptide-based FHE hydrogel (F127/OHA-EPL); DFLs—dermal fibroblasts; ECM—extracellular matrix; MMP—matrix metalloproteinases; SASP—senescence-associated secretory phenotype; IL—interleukin; TNF-α—tumor necrosis factor-α; SC—stratum corneum; SPRY1—sprouty homolog 1; eNAMPT—extracellular nicotinamide phosphoribosyltransferase; VEGF—vascular endothelial growth factor; KGF—keratinocyte growth factor; HF—hair follicle; s.c.—subcutaneous; i.p.—intraperitoneal; i.v.—intravenous; i.d.—intradermal.

## Data Availability

Data sharing not applicable. No new data were created or analyzed in this study.

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
