# Peer review of "Exosomes and Other Extracellular Vesicles with High Therapeutic Potential: Their Applications in Oncology, Neurology, and Dermatology"

_molecules, 2022, doi:10.3390/molecules27041303_

Round 1

Reviewer 1 Report

The authors addressed all my questions in their revised manuscript. I support its acceptance in its current form. 

Author Response

Dear Reviewer,

We sincerely appreciate your efforts put into our contribution. Thank you very much for your earlier comments which made our work better and more complete.

Reviewer 2 Report

In this review article, the authors have reviewed the therapeutic applications of extracellular vesicles (EVs).

Although authors have selected a topic of interest, nevertheless there are several issues which need a significant revision. 

My comments are appended below. 

1. While authors refer therapeutic applications of EVs in the title, it is better to specify which ones?. Otherwise, if authors aim to discuss this for all diseases, then main body should include more diseases beyond cancers and nervous system diseases. This may include the addition of but not limited to; inflammatory diseases, infectious diseases, metabolic diseases, and so. 

2. The content of current review is so much diversified, and this reviewer is afraid that this will distract readers. Better to omit or remove unnecessary information especially those of EV basics which every reader might already know, and keep only the content that is of advanced nature. Otherwise, adding the bit of everything may distract the readers from the main content, and thus the interest. 

3. The abstract provides a superficial yet quick note of EVs, but not the logically presented. I strongly suggest authors to develop the abstract in logical events in the line of current topic. 

4. Since authors aim to deliver the applications of EVs as delivery systems of therapeutic modalities, the alternative delivery system should be introduced in the introduction. Currently, the introduction lacks this. And why EVs are preferred over other delivery systems is also lacking. 

5. In the text authors at some instances use abbreviation EVs, while at other places applied full name ‘extracellular vesicles’. Make the language uniform. Better to abbreviate when first time mentioned in the main body, then later use abbreviation only. 

6. Legends of figure 2 need to be explained. 

7. A major part of content of the current review is cancer, followed by nervous system and then others. Same is with the tables, which cover only few diseases but tables for other diseases are missing. Every topic should be covered in appropriate proportions. Especially when authors aim to cover almost every disease. 

8. Finally, the challenges associated with EV production, isolation, purification, and utilization for therapeutics are not discussed. And recommendations to overcome those challenges should also be presented. 

Author Response

Dear Reviewer,

Thank you for any comments and comments. Please see the attachment - we describe all the corrections made in it.

Reviewer 3 Report

The work by Szwedowicz et al. summarized that possibility and hope as targets for therapy of extracellular vesicles(EVs) were increasing by considering with several reports, and importance as key materials to improve and treat many diseases were convinced based on general reports related to EVs. Authors suggested classification and theory, clinical use, vaccination strategies on several cancers, drug delivery to the nervous system and therapy limitations focused to EVs.  

Although  This manuscripts is well organized and written based on many references with adding concise Tables. So this review must be profitable to many researchers and medicals. And I think that this applied work as manuscripts summarized to carry out medical therapy focused to EVs is pioneering and novel one. And especially  this must be effective one as an article for researchers and medicals in a first step interested to EVs as targets.

Author Response

Dear Reviewer,

We sincerely appreciate your efforts put into our manuscript and giving us another opportunity to improve our work in the best possible way. Thank you for all your informative remarks and comments.

Round 2

Reviewer 2 Report

The authors have addressed my comments and have further improved the manuscript. 

I have no further comments. 

This manuscript is a resubmission of an earlier submission. The following is a list of the peer review reports and author responses from that submission.

Round 1

Reviewer 1 Report

The review examines a wide range of aspects of the therapeutic use of extracellular vesicles, including exosomes. The review considers three main areas of therapeutic application of vesicles, these are cancer treatment, including immunotherapy, therapy of pathologies of the central nervous system, including acute conditions and neurodegeneration, as well as the use of vesicles in cosmetology and dermatology.

The main criticism of the presented work relates to its novelty. The review mainly examines the currently available information, the enumeration of known facts and phenomena, without deep critical analysis. There are a lot of similar reviews on more narrow topics (such as cancer, acute brain damage, Alzheimer's disease, and other neurodegenerations, dermatology, and many others) now. The question is what new this review brings to this field of science. The choice of the key sections of the vesicles application is not very clear. For example, pathologies of other organs (kidney, liver, immune system) are not considered. The available information about the negative effects of the use of vesicles (for example, cases of thrombosis) is not considered. The authors almost do not share the description of extracellular vesicles (very heterogeneous group) and exosomes (which is a subtype with specialized characteristics), so it is difficult to interpret the presented data. The authors do not provide any generalizing schemes illustrating the mechanisms of action of vesicles and/or ways of their use in therapy. The review is overloaded with data, but it does not bring a new understanding to the problem under consideration

Author Response

Dear Reviewer,

 According to your suggestion, we have improved the manuscript. 

Our full answer is in the attached file.

Kind regards, 

Urszula Szwedowicz

Reviewer 2 Report

This manuscript summarises the progress in applying exosomes for disease treatment. This topic is interesting and fits the scope of the journal. The manuscript is well-organized in sound logic. The referee supports its acceptance with minor revisions noted.

  1. Please include one scheme to summarize the main contents of this manuscript.
  2.  Important figures from representative references can be helpful for readers to follow.  
  3. After the Background section, add one section to discuss how the drug/protein/gene therapeutics-encapsulated exosomes are engineered and the common characterization techniques being used. 

Author Response

(The authors gave the same response as above.)

Reviewer 3 Report

antitumoral agents, as immunomodulatory therapies, in the treatment of neurodegenerative diseases, stroke, and skin diseases. The manuscript is valuable because it provides a concise overview of the latest therapeutic uses of exosomes. Several issues, however, must be corrected before acceptance, for instance:

  • The manuscript is difficult to read. The technical data must be organized in tables and include at least two figures.
  • If well the composition of each exosome is briefly described, the administration routes must be indicated particularly for immunomodulators (intravenous? or subcutaneous?). It must be clarified also, if the results were achieved with a single or multiple doses, and the extent of the effects.
  • Another important aspect to be addressed is referred to the techniques employed to load different types of molecules into exosomes. Many described formulations exhibit a high degree of structural complexity, suggesting a very difficult preparation and low reproducibility.  
  • A critical point is a fact that therapeutic exosomes are part of highly personalized treatments, where exosomes are extracted from each patient and modified ad-hoc to treat each disease. This is very important from the point of view of future translation and deserved to be discussed: what about a future production at an industrial scale? Is it necessary? How would the exosomes be preserved? Is it necessary? What about their structural characterization? Would exosomes be considered NBCD?. To that aim, it should be useful to include an explicit comparison with other technologies, such as nanomedicines.
  • Row 425: nanoparticles neither enter cells nor displace by passive diffusion. Please correct.

Author Response

Dear Reviewer,

According to your suggestion, we have improved the manuscript. Thank you for all your suggestions.
Our full answer is in the attached file.

Kind regards, 

Urszula Szwedowicz

Round 2

Reviewer 1 Report

After the revision, the manuscript did not sound any better. Problems remain with the novelty of this review. Other criticism of the reviewer was also not addressed by the authors in the responses and corrections.